# NoLoRA: Nonlinear Low-Rank Adaptation for Parameter-Efficient Fine-Tuning

## Abstract

Low-Rank Adaptation (LoRA) has been widely adopted for parameter-efficient fine-tuning of large language models, as it enables effective adaptation while maintaining efficiency. However, existing LoRA methods are fundamentally linear in nature, relying on the multiplication of two matrices ($B \times A$) for parameter adaptation. This inherently linear structure constrains their expressiveness, rendering them insufficient for capturing higher-order feature interactions and complex nonlinear patterns essential for advanced tasks. Consequently, this linearity becomes a bottleneck that limits further performance improvements. To address this limitation, we propose a nonlinear extension that introduces an activation function and a modulation mechanism into the low-rank adapter (**No**nlinear **Lo**w-**R**ank **A**daptation, **NoLoRA**), enhancing adaptability across diverse tasks. Our design preserves the parameter efficiency and scalability of LoRA while significantly improving representational capacity. Comprehensive experiments on four benchmarks, including commonsense reasoning, natural language understanding, image classification, and mathematical reasoning, demonstrate that our approach achieves consistent and substantial improvements over vanilla LoRA, LoRA's variant and other Parameter-efficient fine-tuning (PEFT) methods, with negligible additional computational overhead. These findings suggest that incorporating lightweight nonlinear structures into parameter-efficient fine-tuning frameworks offers a promising direction for improving the adaptability of large models.

## 1 INTRODUCTION

Pre-trained models serve as the backbone of contemporary machine learning, offering strong generalization capabilities through training on large-scale and heterogeneous corpora. Their success spans across diverse domains, ranging from natural language understanding (Devlin et al., 2019; Liu et al., 2019) and natural language generation (Touvron et al., 2023; AI, 2024) to vision tasks such as image classification (Dosovitskiy et al., 2021). Adapting these powerful models to downstream applications, however, often relies on full fine-tuning, which is computationally expensive and memory-intensive due to the enormous number of parameters involved (Qin et al., 2024). This challenge has motivated the development of parameter-efficient fine-tuning (PEFT) techniques (Ding et al., 2023; Han et al., 2024), which introduce lightweight trainable modules to reduce resource consumption while retaining the adaptability of pre-trained models (Lin et al., 2024).

Among various PEFT approaches, the family of Low-Rank Adaptation (LoRA) methods (Hu et al., 2022; Liu et al., 2024; Song et al., 2024; Büyükakyüz, 2024; Zhao et al., 2024) has emerged as one of the most effective and widely adopted due to its minimal architectural modifications, strong efficiency, and competitive performance. Instead of updating the original model weights directly, LoRA introduces two learnable low-rank matrices whose product approximates the weight update. Since these matrices are significantly smaller than the full weight matrices, LoRA substantially reduces memory usage during fine-tuning.

Despite its widespread adoption, LoRA still faces limitations, particularly in modeling complex weight update patterns. By constraining updates to the product of low-rank matrices, LoRA reduces the parameter search space but at the cost of limited expressiveness, which hinders its ability to capture the intricate patterns required for many downstream tasks (Pan et al., 2024). Specifically, when the rank is small, the approximation often fails to model the complex optimization trajectories

needed for high performance. To narrow the gap with full fine-tuning, higher-rank settings are typically required; however, this increases parameter overhead and undermines the efficiency advantage of LoRA.

Several studies have attempted to alleviate the limitations of LoRA. Recent variants of LoRA, such as MosLoRA (Wu et al., 2025) , improve performance by add a mixer matrix between matrix $A, B$ to mixture the subspaces; Pissa (Meng et al., 2024), initializes LoRA adapters with the principal singular values and singular vectors of weight matrices, thereby achieving faster convergence and better performance than conventional LoRA with the same number of parameters; MiLoRA (Wang et al., 2024), initializes LoRA adapters with the minor singular values and singular vectors of weight matrices, thereby achieving faster convergence and reduce the knowledge forgetting. A novel fine-tune framework named NEAT(Zhong et al., 2025), incorporated a lightweight neural network that models cumulative weight updates as functions of the pre-trained weights, thereby enhancing expressiveness, but at the expense of additional parameters and computation. While these methods demonstrate progress in specific scenarios, they remain fundamentally linear or sacrifice efficiency to gain expressiveness. This leaves a key gap: how to enhance representational capacity while retaining efficiency.

To address this gap, we propose a nonlinear extension of LoRA, termed NoLoRA, which augments the low-rank adaptation module with lightweight nonlinear functions and modulation mechanisms. Unlike vanilla LoRA that relies solely on linear low-rank decompositions, NoLoRA models weight updates through nonlinear transformations, allowing it to capture richer and more complex adaptation patterns. This design enhances the expressive power of the adaptation process while preserving parameter efficiency and incurring only negligible computational overhead. By introducing nonlinear modulation into the update mechanism, NoLoRA achieves improved flexibility in modeling diverse task-specific variations, enabling more effective adaptation than purely linear approaches.

Our contributions are threefold:

- We propose a novel nonlinear LoRA that augments low-rank adaptation with nonlinearities and modulation mechanisms, enhancing expressiveness beyond strictly linear structures.
- We present a simple, efficient approach that avoids complex routing and matrix mapping, enabling effective representation learning and optimal performance across tasks.
- We conduct comprehensive experiments on multiple benchmarks, demonstrating consistent and substantial performance improvements over vanilla LoRA, LoRA's variants and other PEFT methods, with negligible additional overhead.

## 2 RELATED WORKS

**Low-rank adaptation**    Parameter-efficient fine-tuning (PEFT) methods are primarily motivated by the need to mitigate the prohibitive computational and memory costs associated with fully fine-tuning large language models (LLMs). Instead of updating all model parameters, PEFT restricts optimization to a small, carefully designed subset of parameters, thereby preserving efficiency while still enabling effective adaptation to downstream tasks or domain-specific corpora. Within this landscape, low-rank adaptation represents one of the most influential and widely adopted approaches. A pioneering example, LoRA (Hu et al., 2022), factorizes the fine-tuning update into the product of two low-rank matrices, which approximates the weight update without directly altering the original parameters. This design allows LoRA to integrate seamlessly into pre-trained architectures, introducing no additional inference cost and only a minimal number of trainable parameters.

Subsequent work has expanded upon this core idea in various directions. DoRA (Liu et al., 2024) introduces a magnitude-direction decomposition of the original weight matrix and applies LoRA updates exclusively to the directional component, thereby enhancing representational flexibility. MoRA (Jiang et al., 2024) explores another dimension by compressing the input space through predefined functions, transforming it via a square "higher-rank" projection, and subsequently decompressing it, which collectively enables higher-rank adaptation while maintaining computational feasibility. FourierFT (Gao et al., 2024a) replaces the matrix multiplication in LoRA with a Fourier transform, while PiSSA (Meng et al., 2024) and MiLoRA (Wang et al., 2024) update the principal and minor singular components of the weight matrix, respectively.NEAT (Zhong et al., 2025) incorporates a lightweight neural network to model cumulative weight updates as nonlinear functions

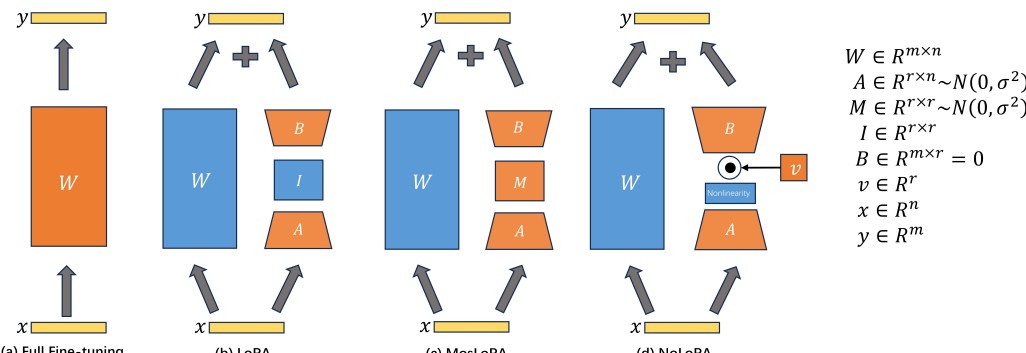

Figure 1: The comparison among Full Fine-tuning, training with LoRA, MosLoRA and NoLoRA. In this visualization, blue modules represent parts of the model whose parameters are frozen during training, while orange modules indicate components that require updates. $I$ is a identify matrix and $v$ is a learnable modulation vector for task-specific adaptation. Nonlinearity means a nonlinear activation function like ReLU. $\odot$ is a hadamard product.

of the pre-trained weights, effectively extending the expressiveness of low-rank parameterization without sacrificing efficiency. Taken together, these innovations underscore the versatility of the low-rank paradigm, while also revealing an ongoing tension between efficiency and expressiveness in PEFT design.

**Other PEFT methods** Beyond low-rank techniques, there are two additional families of PEFT methods have gained significant attention: prompt-based and adapter-based strategies. Prompt-based methods integrate trainable virtual tokens into the model input and restrict optimization solely to these tokens, thereby preserving the frozen parameters of the LLM. For instance, Prompt Tuning Lester et al. (2021) augments the input with task-specific embeddings at the initial layer, whereas P-Tuning (Liu et al., 2022) extends this idea by injecting learnable prompts into every layer of the model. These methods are appealing due to their extremely small parameter footprint, but they are also known to be sensitive to initialization, which can impact convergence and final performance (Wu et al., 2024a). Furthermore, because Transformer models exhibit quadratic complexity with respect to input length (Vaswani et al., 2017), prompt-based approaches can increase inference costs proportionally to the prompt length, potentially limiting their scalability in long-context scenarios.

Adapter-based methods constitute the third major category of PEFT techniques. Unlike prompt-based approaches, which manipulate the input space, adapter-based strategies modify the model architecture by inserting lightweight trainable modules into pre-trained networks. Early works such as Adapters (Houlsby et al., 2019) and Compacter (Karimi Mahabadi et al., 2021) introduced additional linear layers into the LLM backbone, allowing efficient task adaptation while freezing most parameters. Building on this foundation, Parallel Adapters (He et al., 2021) explored the integration of multiple adapters in parallel, thereby improving expressiveness and robustness across tasks. Despite their effectiveness, adapter-based methods inherently alter the architecture of the model during both training and inference, which can lead to additional overhead compared to low-rank or prompt-based alternatives. Nevertheless, they remain an important line of research due to their balance between modularity, extensibility, and performance, especially in multi-task and continual learning settings.

## 3 METHOD

In this section, we first provide a brief overview of LoRA. We then point out a fundamental limitation in its parameter efficiency, which arises from the specific parameterization form of LoRA. To address this issue, we introduce NoLoRA, a novel parameter-efficient fine-tuning (PEFT) method. Our analysis further shows that NoLoRA provably achieves superior parameter efficiency compared to standard LoRA, while maintaining its lightweight design.

## 3.1 PRELIINARY

LoRA(hu2021lora) assumes that weight updates during fine-tuning follow a low-rank structure. Specifically, the update of a pre-trained weight matrix $W_0 \in \mathbb{R}^{m \times n}$ is approximated by the product of two learnable low-rank matrices:

$$W = W_0 + \Delta W = W_0 + BA, \tag{1}$$

where $B \in \mathbb{R}^{m \times r}$, $A \in \mathbb{R}^{r \times n}$, and $r \ll \min(m, n)$. During fine-tuning, the original weights $W_0$ remain frozen, while only the introduced matrices $B$ and $A$ are optimized by solving

$$\min_{B,A} \ L(D_{\text{train}}; W_0 + BA), \tag{2}$$

with $D_{\text{train}}$ denoting the training dataset and $L$ the loss function. Since both $B$ and $A$ are low-rank matrices with far fewer parameters than $W_0$, LoRA significantly reduces memory and computation overhead compared to full fine-tuning.

## 3.2 LIMITATIONS OF LoRA

In the setting of full fine-tuning, the weights of a pre-trained model are typically updated through iterative gradient descent:

$$W_t = W_{t-1} - \eta \nabla_{W_{t-1}} \mathcal{L}, \tag{3}$$

where $\eta$ is the learning rate. After $t$ iterations, the cumulative change in the weights can be expressed as:

$$\Delta W = W_t - W_0, \tag{4}$$

This cumulative update $\Delta W$ characterizes the adaptation dynamics of the model on downstream tasks, which are often highly nonlinear and complex. Therefore, capturing such dynamics with sufficient expressiveness is crucial for effective fine-tuning.

LoRA approximates the weight update as a low-rank decomposition:

$$\Delta W \approx BA. \tag{5}$$

where $A$ and $B$ are trainable low-rank matrices. Although this design substantially reduces the number of trainable parameters, the strict linear form constrains the expressive capacity of the learned update. Under low-rank settings, the approximation often struggles to capture the nonlinear dependencies and intricate optimization trajectories required by many downstream tasks, leading to suboptimal performance. Increasing the rank can partially mitigate this limitation, but at the expense of additional parameter overhead, which undermines the efficiency advantage of LoRA.

## 3.3 NONLINEAR EXTENSION OF LoRA

A key limitation of standard LoRA lies in its strictly linear low-rank parameterization, which restricts the expressiveness of weight updates and limits adaptation to complex downstream tasks. Specifically, the linear product $BA$ can only capture simple weight transformations, and fails to model higher-order dependencies or nonlinear interactions. This limitation is particularly pronounced under low-rank settings, as the parameter space is tightly constrained and cannot approximate the complex dynamics of full fine-tuning. Although increasing the rank can partially mitigate this issue, it significantly increases the parameter count and undermines LoRA's parameter efficiency.

As shown in figure 1, to address this limitation, we propose a nonlinear extension of LoRA, which enhances representational capacity while preserving parameter efficiency by incorporating a nonlinear activation function and a task-specific modulation vector. Formally, the weight update for a pre-trained layer is defined as:

$$\Delta W(x) = B \left( v \odot f(Ax) \right), \tag{6}$$

where $A$ and $B$ are low-rank matrices, $f(\cdot)$ denotes the activation function such as ReLU, and $v$ is a learnable modulation vector for task-specific adaptation. Compared to LoRA's linear product $BA$, this design introduces nonlinearity along the update path, enabling the weight update to capture more complex interactions while maintaining a small parameter footprint. The modulation vector

$v$ provides dynamic scaling of features, allowing the model to adaptively adjust weight updates according to the downstream task. By combining the nonlinear activation and modulation mechanism, the method significantly enhances task adaptability while preserving LoRA's memory and computational efficiency, and more closely approximates the expressive power of full fine-tuning.

During fine-tuning, the original pre-trained weights $W_0$ are frozen, and only the parameters $A, B, v$ are updated by minimizing the task-specific loss:

$$\min_{A,B,v} \mathcal{L}(\mathcal{D}_{\text{train}}; W_0 + B\, v \,\odot\, f(Ax))). \tag{7}$$

Despite the addition of nonlinearity and the modulation mechanism, the extra parameters remain far fewer than full fine-tuning. The parameter count of vanilla LoRA is $A \in \mathbb{R}^{r \times n}, B \in \mathbb{R}^{m \times r}$, and our method only adds $v \in \mathbb{R}^r$, maintaining overall parameter efficiency. Experimental results show that our method outperforms standard LoRA and its variants across multiple downstream tasks, demonstrating the effectiveness of the nonlinear extension and modulation mechanism in parameter-efficient fine-tuning. Despite the addition of nonlinearity and the modulation mechanism, the extra parameters remain far fewer than full fine-tuning.

In summary, by introducing nonlinear mapping and modulation, our method can express more complex interactions, approximating the performance of full fine-tuning. Formally, for any smooth target weight update $\Delta W$, there exists a set of parameters $A, B, v$ such that $B(v \odot f(Ax))$ can approximate $\Delta W$ to arbitrary precision. This analysis illustrates the improved expressiveness of our method and provides theoretical support for the empirical results, validating the effectiveness of the nonlinear extension and modulation mechanism in parameter-efficient fine-tuning.

# 4 EXPERIMENT

In the experiments, we evaluate the proposed NoLoRA and answer the following questions: RQ1) How does NoLoRA compare to widely adopted PEFT methods on NLP tasks? RQ2) How does NoLoRA compare to widely adopted PEFT methods on vision tasks?

## 4.1 BENCHMARKS AND EXPERIMENT SETUPS

We experiment NoLoRA on datasets from four representative benchmarks:

**Commonsense Reasoning.** We conduct experiments on Commonsense170K, a benchmark proposed by (Hu et al., 2023) that unifies eight commonsense reasoning sub-tasks with standardized training and testing splits. This collection covers diverse aspects of commonsense reasoning, including BoolQ (binary yes/no question answering) Clark et al. (2019), PIQA (physical commonsense reasoning)(Bisk et al., 2020), SIQA (social interaction reasoning) (Sap et al., 2019), HellaSwag (commonsense natural language inference) (Zellers et al., 2019), WinoGrande (fill-in-the-blank coreference resolution) (Sakaguchi et al., 2020), ARC-e and ARC-c (easy and challenge subsets of multiple-choice science QA) Clark et al. (2018), and OBQA (multi-step reasoning) (Mihaylov et al., 2018). In total, the benchmark provides 170,420 query–answer pairs for fine-tuning LLMs, along with 120 randomly sampled instances held out for validation. By integrating datasets with complementary reasoning challenges, Commonsense170K enables a comprehensive evaluation of model generalization across different forms of commonsense.

**Natural Language Understanding.** Natural Language Understanding (NLU) experiments are conducted on eight datasets from the GLUE benchmark (Wang et al., 2018), which cover a wide spectrum of linguistic phenomena including entailment, paraphrase detection, sentiment analysis, and sentence similarity. Specifically, GLUE comprises tasks such as MNLI (multi-genre natural language inference), QNLI (question–answer entailment), RTE (recognizing textual entailment), SST-2 (sentiment classification), MRPC (paraphrase identification), QQP (duplicate question detection), CoLA (linguistic acceptability), and STS-B (semantic textual similarity). To ensure comparability with prior work, we adopt the evaluation metrics and experimental protocols established in (Gao et al., 2024a) and (Wu et al., 2024b), where accuracy is reported for classification tasks, Matthew's correlation for CoLA, and Pearson/Spearman correlations for STS-B.

**Image Classification.** Image Classification is evaluated on eight benchmarks following (Gao et al., 2024b): Oxford-Pets (Parkhi et al., 2012), CIFAR10 (Krizhevsky, 2009), DTD (Cimpoi et al., 2014),

EuroSAT (Helber et al., 2019), RESISC45 (Cheng et al., 2017), StanfordCars (Krause et al., 2013), FGVC (Maji et al., 2013), and CIFAR100 (Krizhevsky, 2009). The first five datasets represent small label spaces (10–47 categories), while the latter three involve large, fine-grained label spaces (100+ categories). This setup enables evaluation across both general and fine-grained classification scenarios.

**Arithmetic Understanding.** For this task, we employ MetaMath (Yu et al., 2023) as the training corpus and GSM8K (Cobbe et al., 2021) as the test dataset.Models need to generate correct answers, and accuracy is used as the evaluation metric.

**Baselines methods** are chosen on a task basis. On each task, NoLoRA is compared with representative baselines from the corresponding domain.Further details regarding the dataset, baselines and hyperparametes are provided in the Appendix.

## 4.2 RESULTS ON COMMONSENSE REASONING TASKS

To address RQ1, we evaluate our method on eight commonsense reasoning datasets under the LLaMA3-8B backbone. As shown in Table 1, our approach achieves strong results across different configurations, with an average accuracy of **85.2%** when $r = 8$, **85.6%** when $r = 16$, and **85.8%** when $r = 32$. The latter establishes a new state-of-the-art among parameter-efficient fine-tuning (PEFT) methods. Compared with existing baselines, our best configuration ($r = 32$) yields consistent and substantial improvements: **+5.0%** over LoRA, **+10.4%** over PiSSA, **+4.0%** over MiLoRA, **+3.3%** over NEAT, and **+0.9%** over MosLoRA under the same rank setting.

In terms of per-task performance, our method consistently delivers improvements across all benchmarks when $r = 32$. Specifically, it achieves **ARC-e (+3.7%)**, **OBQA (+3.4%)**, **SIQA (+1.0%)**, **ARC-c (+4.5%)**, **WinoG (+1.0%)**, **PIQA (+5.4%)**, **BoolQ (+2.9%)**, and **HellaS (+5.1%)**. These results demonstrate that our approach achieves consistent gains across diverse tasks, particularly excelling in challenging benchmarks such as ARC-c, PIQA, BoolQ, and HellaS.

Overall, our method delivers the best or comparable results on all benchmarks, consistently surpassing LoRA, PiSSA, MiLoRA, MosLoRA, and NEAT. These findings highlight the effectiveness of our approach in overcoming the limitations of existing PEFT methods. By introducing a more expressive yet parameter-efficient adaptation mechanism, our method achieves stronger and more stable performance across diverse commonsense reasoning tasks, especially in complex scenarios.

Table 1: Commonsense Reasoning performance of NoLoRA and PEFT baselines on LLaMA3-8B. Results marked with "+" are taken from DoRA, and those marked with "*" are taken from Milora. Best results are in bold. "AVG" means the average accuracy of all datasets.

| PEFT | %Param | ARC-e | OBQA | SIQA | ARC-c | WinoG | PIQA | BoolQ | HellaS | Avg. |
|------|--------|-------|------|------|-------|-------|------|-------|--------|------|
| LoRA$^+_{r=32}$ | 0.7002 | 84.2 | 79.0 | 79.9 | 71.2 | 84.3 | 85.2 | 70.8 | 91.7 | 80.8 |
| PiSSA*$_{r=32}$ | 0.7002 | 77.7 | 74.6 | 77.2 | 63.2 | 78.9 | 81.1 | 67.1 | 83.6 | 75.4 |
| MiLoRA*$_{r=32}$ | 0.7002 | 86.8 | 81.9 | 77.2 | 75.5 | 85.6 | 86.7 | 68.8 | 92.9 | 81.9 |
| MosLoRA$_{r=16}$ | 0.3545 | 90.1 | 84.2 | 79.8 | 79.8 | 86.6 | 88.6 | 74.6 | 95.1 | 84.9 |
| NEAT$_{r=32}$ | 0.7001 | 87.0 | 83.0 | 79.6 | 77.2 | 85.6 | 83.8 | 72.7 | 90.8 | 82.5 |
| NoLoRA$_{r=4}$ | 0.0880 | 90.7 | 83.2 | 81.0 | 77.6 | 84.8 | 88.5 | 71.6 | 95.6 | 84.1 |
| NoLoRA$_{r=8}$ | 0.1759 | 90 | 84.6 | 81.5 | 80 | 87.1 | 88.4 | 74.3 | 95.3 | 85.2 |
| NoLoRA$_{r=16}$ | 0.3513 | 90.8 | 85.2 | 80.9 | 79.8 | 88.0 | 89.1 | 75.3 | 95.5 | 85.6 |
| NoLoRA$_{r=32}$ | 0.7004 | 90.7 | 86.4 | 80.6 | 81.7 | 86.6 | 89.2 | 75.6 | 95.9 | **85.8** |

## 4.3 RESULTS ON NLU TASKS

To address RQ2, We further evaluate our proposed method on the GLUE benchmark to verify its effectiveness on natural language understanding tasks to address RQ1. As shown in Table 2, our method achieves the highest average score of **86.5%**, outperforming all compared PEFT methods. Compared to the widely adopted LoRA baseline, our method delivers a consistent improvement of

Table 2: Performance comparison on the GLUE benchmark under the RoBERTa-base backbone. Results marked with "*" are taken from (Wu et al., 2024a).Results marked with "+" are taken from (Gao et al., 2024a) The best results are highlighted in **bold**."AVG" means the average accuracy of all datasets.

| PEFT | %Param | MNLI | SST-2 | MRPC | CoLA | QNLI | QQP | RTE | STS-B | AVG |
|---|---|---|---|---|---|---|---|---|---|---|
| FFT* | 100% | 87.3 | 94.4 | 87.9 | 62.4 | 92.5 | 91.7 | 78.3 | 90.6 | 85.6 |
| Adapter* | 0.318% | 87.0 | 93.3 | 88.4 | 60.9 | 92.5 | 90.5 | 76.5 | 90.5 | 85.0 |
| LoRA* | 0.239% | 86.6 | 93.9 | 88.7 | 59.7 | 92.6 | 90.4 | 75.3 | 90.3 | 84.7 |
| Adapter$^{FNN}$* | 0.239% | 87.1 | 93.8 | 88.8 | 58.5 | 92.0 | 90.2 | 77.7 | 90.4 | 85.1 |
| BitFit* | 0.080% | 84.7 | 94.0 | 88.0 | 54.0 | 91.0 | 87.3 | 69.8 | 89.5 | 82.3 |
| RED* | 0.016% | 83.9 | 93.9 | 89.2 | 61.0 | 90.7 | 87.7 | 78.0 | 90.4 | 84.4 |
| FourierFT$^+$ | 0.019% | 84.7 | 94.3 | 90.0 | 63.8 | 92.2 | 88.8 | 79.1 | 90.8 | 85.5 |
| DiReFT* | 0.015% | 82.5 | 92.6 | 88.3 | 58.6 | 91.3 | 86.6 | 76.4 | 89.3 | 83.2 |
| LoReFT* | 0.019% | 83.1 | 93.4 | 89.2 | 60.4 | 92.1 | 87.4 | 79.0 | 90.0 | 84.2 |
| NEAT | 0.241% | 86.9 | 94.5 | 88.2 | 64.6 | 92.8 | 90.3 | 78 | 91 | 85.8 |
| NoLoRA | 0.239% | 86.8 | 95.0 | 89.5 | 67.0 | 92.7 | 90.5 | 80.5 | 90.3 | **86.5** |

Table 3: Performance comparison on eight image classification datasets. All methods use the ViT-base. The best results are highlighted in **bold**."AVG" means the average accuracy of all datasets.Results marked with "*" are taken from (Gao et al., 2024a)

| Method | Params(M) | OxfordPets | Cars | CIFAR10 | DTD | EuroSAT | FGVC | RESISC45 | CIFAR100 | AVG |
|---|---|---|---|---|---|---|---|---|---|---|
| FFT* | 85.8M | 93.14 | 79.78 | 98.92 | 77.68 | 99.05 | 54.84 | 96.13 | 92.38 | 86.49 |
| LP* | - | 90.28 | 25.76 | 96.41 | 69.77 | 88.72 | 17.44 | 74.22 | 84.28 | 68.36 |
| LoRA* | 581K | 93.19 | 45.38 | 98.78 | 74.95 | 98.44 | 25.16 | 92.70 | 92.02 | 77.58 |
| FourierFT* | 239K | 93.05 | 56.36 | 98.69 | 77.30 | 98.78 | 32.44 | 94.26 | 91.45 | 80.29 |
| NEAT | 263K | 93.62 | 80.21 | 98.78 | 79.61 | 98.85 | 52.93 | 94.71 | 92.02 | 86.34 |
| NoLoRA | 295K | 96.47 | 81.47 | 98.94 | 80.22 | 98.83 | 54.19 | 98.81 | 92.26 | **87.27** |

**+1.8%** points, and also surpasses more recent approaches such as Adapter$^{FNN}$(**+1.4%**), FourierFT (**+1.0%**), NEAT (**+0.7%**) and even better than FFT(**+0.9%**).

Beyond the overall average, our method demonstrates robust improvements on individual tasks. For example, it achieves **67.0**% on CoLA, significantly higher than LoRA (59.7%) and NEAT (64.6); **92.7%** on QNLI, the best among all methods; and **95.0%** on SST-2, surpass the strongest baseline NEAT (94.5%). These results highlight that our method not only improves overall performance but also consistently advances the state of the art across diverse tasks such as entailment, sentiment analysis, paraphrase detection, and linguistic acceptability.

Overall, the strong results on GLUE confirm the generality of our approach: by introducing a more expressive yet parameter-efficient adaptation mechanism, our method continues to push the state of the art in natural language understanding.

## 4.4 RESULTS ON IMAGE CLASSIFICATION TASKS

We further evaluate our method on eight diverse image classification datasets to assess its generalization ability in the vision domain. As shown in Table 3, our method achieves the highest average accuracy of **87.27%**, outperforming all baselines. Compared to LoRA, our approach provides a substantial gain of **+9.7%**, and also surpasses FourierFT (**+7.0%**), NEAT (**+0.9%**) and even better than FFT(**+0.78%**).

In particular, our method sets new records on multiple datasets, such as **96.47%** on OxfordPets, **81.47%** on StanfordCars, and **98.81%** on RESISC45, significantly exceeding prior PEFT approaches. These results demonstrate that our method not only improves average accuracy but also consistently advances performance across diverse domains, highlighting its strong adaptability to vision tasks.

Table 4: Arithmetic Reasoning performance on LLaMA2-7B. The best result is highlighted in **bold**.Results marked with "*" are taken from (Zhong et al., 2025)

| Method | %Param | GSM8K |
|---|---|---|
| FFT* | 100% | 66.5 |
| LoRA*$_{r=64}$ | 1.662% | 60.58 |
| PiSSA*$_{r=64}$ | 1.662% | 58.23 |
| MiLoRA*$_{r=64}$ | 1.662% | 63.53 |
| NEAT*$_{r=64}$ | 1.662% | 65.05 |
| NoLoRA$_{r=16}$ | 0.417% | 63.40 |
| NoLoRA$_{r=32}$ | 0.834% | 65.96 |
| NoLoRA$_{r=64}$ | 1.668% | **66.87** |

## 4.5 RESULTS ON ARITHMETIC REASONING TASK

We evaluate our method on the GSM8K dataset to examine its effectiveness on mathematical reasoning tasks. As shown in Table 4, our method achieves the highest accuracy of **66.87%**, surpassing all compared PEFT baselines. Compared with LoRA, it improves by **+6.29%**, and also outperforms PiSSA (**+8.64%**), MiLoRA (**+3.34%**), NEAT (**+1.82%**) and even better than FFT(**+0.37%**). Especially, When r=16,it goes beyond LoRA(r=64) and PISSA(r=64);when r=32, it goes beyond all the PEFT methods.These results demonstrate that our approach effectively enhances performance in complex reasoning tasks while maintaining parameter efficiency.

## 5 ABLATION STUDY

### 5.1 EFFECT OF DIFFERENT ACTIVATION FUNCTIONS

We study the impact of different activation functions on performance across eight commonsense reasoning datasets. Table 5 summarizes the results for ReLU, Tanh, and GELU variants within our method removed task-special modulation vector when r = 16. It shows that using ReLU achieves the highest average accuracy of **85.1%**, slightly outperforming GELU (**84.5%**) and Tanh (**84.7%**).

Although all three activation functions yield competitive performance, ReLU consistently provides better results on most individual datasets, including ARC-e (**91.0%**), PIQA (**89.1%**), and Hellaswag (**95.6%**). These observations suggest that the choice of activation function can influence fine-grained reasoning capabilities, with ReLU being slightly more effective for our adaptation mechanism.

Table 5: Impact of different activation functions on eight commonsense reasoning datasets. The best results are highlighted in **bold**."AVG" means the average accuracy of all datasets.

| Activation | ARC-e | OBQA | SIQA | ARC-c | WinoG | PIQA | BoolQ | HellaS | Avg. |
|---|---|---|---|---|---|---|---|---|---|
| ReLU | 91.0 | 85.6 | 80.2 | 80.3 | 84.9 | 89.1 | 74.0 | 95.6 | **85.1** |
| Tanh | 89.9 | 85.6 | 79.9 | 80.0 | 84.4 | 88.3 | 74.2 | 95.3 | 84.7 |
| GELU | 90.1 | 86.0 | 80.6 | 79.2 | 85.8 | 88.2 | 70.8 | 95.6 | 84.5 |

### 5.2 EFFECT OF MODULATION VECTOR

We investigate the impact of introducing a modulation vector $v$ in combination with different activation functions on performance across eight commonsense reasoning datasets. Table 6 presents the results for ReLU, Tanh, and GELU variants with and without the modulation vector.

The results demonstrate that adding the modulation vector consistently improves performance. For instance, with ReLU at rank 16, including the modulation vector (a_ReLU_v_b$_{r=16}$) increases the average accuracy from **85.1%** to **85.4%**. Similarly, for GELU at rank 16, the modulation vector (a_GELU_v_b$_{r=16}$) boosts average accuracy from **84.5%** to **85.6%**. Using Tanh at rank 16, the

Table 6: Impact of the modulation vector $v$ on commonsense reasoning datasets. The best results are highlighted in **bold**.

| Variant | ARC-e | OBQA | SIQA | ARC-c | WinoG | PIQA | BoolQ | HellaS | Avg. |
|---|---|---|---|---|---|---|---|---|---|
| a_Tanh_b$_{r=16}$ | 89.9 | 85.6 | 79.9 | 80.0 | 84.4 | 88.3 | 74.2 | 95.3 | 84.7 |
| a_Tanh_v_b$_{r=16}$ | 90.3 | 85.5 | 80.3 | 79.5 | 86.0 | 88.6 | 74.2 | 95.5 | 85.0 |
| a_GELU_b$_{r=16}$ | 90.1 | 86.0 | 80.6 | 79.2 | 85.8 | 88.2 | 70.8 | 95.6 | 84.5 |
| a_GELU_v_b$_{r=16}$ | 90.8 | 85.2 | 80.9 | 79.8 | 88.0 | 89.1 | 75.3 | 95.5 | 85.6 |
| a_ReLU_b$_{r=16}$ | 91.0 | 85.6 | 80.2 | 80.3 | 84.9 | 89.1 | 74.0 | 95.6 | 85.1 |
| a_ReLU_v_b$_{r=16}$ | 90.6 | 85.8 | 80.9 | 81.0 | 86.0 | 88.7 | 74.6 | 95.8 | 85.4 |

modulation vector (`a_Tanh_v_b`$_{r=16}$) raises the average accuracy from **84.7%** to **85.0%**. These observations confirm that the modulation vector effectively enhances the adaptation capability of our method without introducing significant additional parameters.

### 5.3 EFFECT OF RANK

To better understand the capacity–efficiency trade-off of our method, we evaluate different rank settings ($r = 4, 8, 16, 32$) across multiple activation functions and with or without modulation vectors on commonsense reasoning datasets and take their average. Results are summarized in Figure 2.

We observe that increasing the rank consistently improves performance up to $r = 32$, confirming that higher-rank updates provide richer representational capacity. For instance, GELU with modulation improves from 84.6% at $r = 4$ to 85.4% at $r = 8$, and further to 85.8% at $r = 32$. A similar trend is observed for ReLU, which achieves the overall best result (**85.8%**) at $r = 32$ with modulation. Tanh also follows this pattern, reaching its peak at $r = 32$ with 85.5%.

It is interesting that modulation vectors bring substantial gains with GELU, especially under low ranks, while their impact on Tanh and ReLU is relatively minor. This suggests that the expressive advantage of modulation vectors is more pronounced when the base activation function has richer nonlinear dynamics (e.g., GELU) and when the model operates under a restricted rank budget.

Overall, these findings highlight that rank plays a central role in balancing expressiveness and efficiency: higher ranks lead to more stable improvements, while modulation vectors act as an efficient complement when rank is limited.

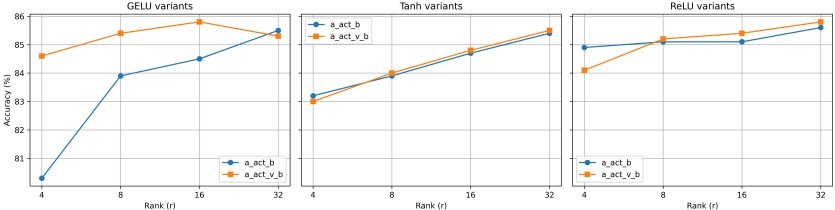

Figure 2: Average accuracy (%) of commonsense reasoning tasks for different activation functions (GELU, Tanh, ReLU) with varying rank values ($r$) and modulation vectors.

## 6 CONCLUSION

In this work, we present NoLoRA, a nonlinear extension of low-rank adaptation that enhances the expressiveness of parameter-efficient fine-tuning. By introducing nonlinear activations and task-specific modulation vectors, NoLoRA overcomes the linear limitations of LoRA while preserving efficiency. Experiments show its superiority on commonsense reasoning, natural language understanding, image classification and arithmetic reasoning benchmarks. In the future, we plan to extend NoLoRA to multimodal tasks, larger models, and integration with other PEFT strategies.

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

APPENDIX

# A BASELINE

To provide a comprehensive comparison, we briefly summarize several representative parameter-efficient fine-tuning (PEFT) approaches that are closely related to our work. These methods vary in their design choices, ranging from low-rank adaptations and adapter modules to lightweight bias tuning and frequency-domain parameterization. Below we outline the core ideas of each method:

**LORA.** LoRA (Hu et al., 2022) inserts trainable low-rank matrices into the frozen pre-trained weights. This technique achieves efficient fine-tuning with fewer trainable parameters and less GPU memory usage.

**Pissa.** PiSSA (Meng et al., 2024) improves upon LoRA by initializing the adapter with principal singular values and vectors, optimizing the key components while freezing the "noisy" ones. This method leads to faster convergence and better performance than LoRA.

**MiLoRA.** MiLoRA (Wang et al., 2024) improves upon LoRA by initializing the adapter with minor singular values and vectors, optimizing the minor components while freezing the principal components. This method leads to faster convergence and better performance than LoRA. Meanwhile, it also reduce the knowledge forgetting.

**MosLoRA.** MosLoRA (Wu et al., 2025) introducing a learnable mixer to combine multiple low-rank subspaces, thereby improving adaptability and representational capacity.

**NEAT.** NEAT incorporates a lightweight neural network into the adaptation process. Unlike LoRA, which approximates weight updates linearly through low-rank decomposition, NEAT models cumulative weight updates as explicit functions of the pre-trained model's original weights.

**Adapter.** Adapter Houlsby et al. (2019) inserts small bottleneck networks into each layer of a pretrained model and trains only these additional modules, enabling parameter-efficient adaptation to downstream tasks.

**BitFit.** BitFit (Zaken et al., 2021) updates only the bias terms of a pretrained model while freezing all other parameters, providing an extremely lightweight yet surprisingly effective parameter-efficient fine-tuning method.

**RED.** RED (Wu et al., 2024a) modifies intermediate representations via lightweight scaling and biasing layers, enabling parameter-efficient adaptation without updating most of the pretrained model.

**FourierFT.** FourierFT (Gao et al., 2024b) learns a small number of frequency-domain coefficients of weight updates via discrete Fourier transform, enabling parameter-efficient adaptation while maintaining model performance.

**ReFT.** ReFT (Wu et al., 2024b) can be implemented in two ways: Direct ReFT (DiReFT), which adds trainable vectors directly to hidden representations, and Low-Rank ReFT (LoReFT), which constrains these updates to a low-rank subspace for parameter-efficient adaptation.

**LP.** LP(Linear probing) is a simple fine-tuning strategy where the pretrained model is frozen and only a lightweight linear classifier is trained on top of its representations.

# B DADASETS

## B.1 COMMONSENSE REASONING

For the commonsense reasoning task, we conduct evaluation on eight widely used benchmark datasets: BoolQ, PIQA, SIQA, HellaSwag, WinoGrande, ARC-e, ARC-c, and OBQA. These datasets collectively cover diverse aspects of commonsense knowledge, ranging from binary question answering and physical or social reasoning to narrative completion, coreference resolution, and scientific problem solving. By including both general-purpose and domain-specific benchmarks, this suite provides a broad and challenging testbed for evaluating the robustness and generalization ability of our approach. The detailed statistics and characteristics of each dataset are summarized in Table 7.

Table 7: Detailed information of commonsense reasoning task.

| Dataset | #Class | #Train | #Dev | #Test |
|---|---|---|---|---|
| BoolQ | Binary classification | 9,427 | 3,270 | 3,245 |
| PIQA | Binary classification | 16,113 | 1,838 | 3,000 |
| SIQA | Ternary classification | 33,410 | 1,954 | 2,224 |
| HellaSwag | Quaternary classification | 39,905 | 10,042 | 10,003 |
| WinoGrande | Binary classification | 40,398 | 1,267 | 1,767 |
| ARC-e | Quaternary classification | 2,251 | 570 | 2,376 |
| ARC-c | Quaternary classification | 1,119 | 229 | 1,172 |
| OBQA | Quaternary classification | 4,957 | 500 | 500 |

## B.2 NATURAL LANGUAGE UBDERSTANDING

The GLUE benchmark comprises 8 NLP datasets: MNLI, SST-2, MRPC, CoLA, QNLI, QQP, RTE, and STS-B, covering tasks such as inference, sentiment analysis, paraphrase detection, linguistic acceptability, question-answering, and textual similarity. STS-B is a regression task, while all other tasks are either single-sentence or sentence-pair classification tasks. We provide detailed information about them in Table 8.

Table 8: Detailed information of the GLUE benchmark.

| Corpus | Task | Metric | # Train | # Val | # Test | # Labels |
|---|---|---|---|---|---|---|
| | | Single-Sentence Tasks | | | | |
| CoLA | Acceptability | Matthews Corr. | 8.55k | 1.04k | 1.06k | 2 |
| SST-2 | Sentiment | Accuracy | 67.3k | 872 | 1.82k | 2 |
| | | Similarity and Paraphrase Tasks | | | | |
| MRPC | Paraphrase | Accuracy / F1 | 3.67k | 408 | 1.73k | 2 |
| STS-B | Sentence similarity | Pearson / Spearman Corr. | 5.75k | 1.5k | 1.38k | 1 |
| QQP | Paraphrase | Accuracy / F1 | 364k | 40.4k | 391k | 2 |
| | | Inference Tasks | | | | |
| MNLI | NLI | Accuracy | 393k | 19.65k | 19.65k | 3 |
| QNLI | QA / NLI | Accuracy | 105k | 5.46k | 5.46k | 2 |
| RTE | NLI | Accuracy | 2.49k | 277 | 3k | 2 |

## B.3 IMAGE CLASSIFICATION

For image classification, we provide detailed information about the used datasets in Table 9.

Table 9: Detailed information of image classification tasks.

| Dataset | #Class | #Train | #Val | #Test | Rescaled resolution |
|---|---|---|---|---|---|
| OxfordPets | 37 | 3,312 | 368 | 3,669 | |
| StandfordCars | 196 | 7,329 | 815 | 8,041 | |
| CIFAR10 | 10 | 45,000 | 5,000 | 10,000 | |
| DTD | 47 | 4,060 | 452 | 1,128 | $224 \times 224$ |
| EuroSAT | 10 | 16,200 | 5,400 | 5,400 | |
| FGVC | 100 | 3,000 | 334 | 3,333 | |
| RESISC45 | 45 | 18,900 | 6,300 | 6,300 | |
| CIFAR100 | 100 | 45,000 | 5,000 | 10,000 | |

## B.4 ARITHMETIC REASONING

Detailed information for arithmetic reasoning task is provided in Table 10. GSM8K consists of high quality grade school math problems, typically free-form answers.

Table 10: Detailed information of arithmetic reasoning task.

| Dataset | #Train | #Dev | #Test |
|---------|--------|------|-------|
| GSM8K | 7,473 | 1,319 | 1,319 |

## C  HYPERPARAMETERS

### C.1  COMMONSENSE REASONING

We provide hyperparameters settings of NEAT, MosLoRA and NoLoRA for commonsense reasoning task in Table 11.

Table 11: Hyperparameters of commonsense reasoning for NEAT, MosLoRA and NoLoRA.

| Hyperparameter | NEAT | MosLoRA | NoLoRA |
|----------------|------|---------|--------|
| Optimizer | | AdamW | |
| Dropout | | 0.05 | |
| Batch size | | 16 | |
| Target module | | q,k,v,up,down | |
| Epochs | | 3 | |
| Rank $r$ | 32 | 16 | 4, 8, 16, 32 |
| $\alpha$ | 32 | 32 | $2r$ |
| Learning rate | 3e-4 | 3e-5 | 1e-4 |
| Warmup steps | | 100 | |

### C.2  NATURAL LANGUAGE UBDERSTANDING

We provide used hyper-parameters for NEAT and NoLoRA in natural language understanding on the GLUE benchmark in Table 12.

Table 12: Hyperparameter settings across GLUE benchmark for NEAT and NoLoRA.

| Hyperparameter | STS-B | RTE | MRPC | CoLA | SST-2 | QNLI | MNLI | QQP |
|----------------|-------|-----|------|------|-------|------|------|-----|
| Optimizer | | | | AdamW | | | | |
| LR Schedule | | | | Linear | | | | |
| Learning Rate (Head) | 5e-3 | 5e-3 | 5e-3 | 1e-3 | 5e-3 | 1e-3 | 5e-3 | 5e-3 |
| Learning Rate (NEAT) | 5e-3 | 5e-3 | 5e-3 | 1e-3 | 5e-3 | 1e-3 | 5e-3 | 5e-3 |
| Learning Rate (NoLoRA) | 5e-3 | 5e-3 | 5e-3 | 1e-3 | 5e-3 | 1e-3 | 5e-3 | 5e-3 |
| Scaling | 0.1 | 0.01 | 0.01 | 0.1 | 0.01 | 0.01 | 0.01 | 0.01 |
| Max Seq. Len | 512 | 512 | 512 | 512 | 512 | 512 | 512 | 512 |
| Batch Size | 64 | 32 | 64 | 64 | 32 | 32 | 32 | 64 |

### C.3  IMAGE CLASSIFICATION

Hyperparameters for NEAT are provided in Table 13. We tune the classification head and the backbone separately and provide detailed settings for each dataset. The scaling factor s is set to 1.0. The rank r for MHSA is set to 7 in the QV-setting.

### C.4  ARITHMETIC REASONING

We provide hyperparameters settings of NEAT and NoLoRA for arithmetic reasoning task in Table 14. We follow the hyper-parameters settings in (Wang et al., 2024). We limit all samples to a maximum of 2048 tokens. For evaluation, we set a maximum token number of 256 on GSM8K (Cobbe et al., 2021) dataset.

Table 13: Hyperparameters for image classification for NEAT and NoLoRA.

| Hyperparameter | OxfordPets | StanfordCars | CIFAR10 | DTD | EuroSAT | FGVC | RESISC45 | CIFAR100 |
|---|---|---|---|---|---|---|---|---|
| Epochs | | | | 10 | | | | |
| Optimizer | | | | AdamW | | | | |
| LR Schedule | | | | Linear | | | | |
| Weight Decay | 8e-4 | 4e-5 | 9e-5 | 7e-5 | 3e-4 | 7e-5 | 3e-4 | 1e-4 |
| Learning Rate (Head) | 5e-3 | 1e-2 | 5e-3 | 1e-2 | 5e-3 | 1e-2 | 5e-3 | 5e-3 |
| Learning Rate (NEAT) | 5e-3 | 1e-2 | 5e-3 | 1e-2 | 5e-3 | 1e-2 | 5e-3 | 5e-3 |
| Learning Rate (NoLoRA) | 5e-3 | 1e-2 | 5e-3 | 1e-2 | 5e-3 | 1e-2 | 5e-3 | 5e-3 |

Table 14: Hyperparameters of arithmetic reasoning for NoLoRA.

| Hyperparameter | NoLoRA |
|---|---|
| Optimizer | AdamW |
| Dropout | 0.05 |
| Batch size | 16 |
| Target module | q,k,v,up,down |
| Warmup steps | 100 |
| Epochs | 3 |
| Rank $r$ | 16, 32, 64 |
| $\alpha$ | $2r$ |
| Learning rate | 1e-4 |

## C.5 ABLATION STUDY

We provide hyperparameters settings of NoLoRA for ablation study on commensense reasoning tasks in Table 15. $\text{NoLoRA}_{(\cdot)}$ means using corresponding activation function between matrix $B$ and $A$.

Table 15: Hyperparameters of commonsense reasoning for NoLoRA's ablation study.

| Hyperparameter | $\text{NoLoRA}_{\text{GELU}}$ | $\text{NoLoRA}_{\text{ReLU}}$ | $\text{NoLoRA}_{\text{Tanh}}$ |
|---|---|---|---|
| Optimizer | | AdamW | |
| Dropout | | 0.05 | |
| Batch size | | 16 | |
| Target module | | q,k,v,up,down | |
| Epochs | | 3 | |
| Rank $r$ | | 4, 8, 16, 32 | |
| $\alpha$ | | $2r$ | |
| Warmup steps | | 100 | |
| Learning rate | 1e-4 | 3e-4 | 3e-5 |

## D   THE USE OF LARGE LANGUAGE MODELS

This manuscript has been polished with the assistance of an LLM, which was used solely for language refinement and not for ideation or substantive writing.

