# OpenReview forum: "NoLoRA: Nonlinear Low-Rank Adaptation for Parameter-Efficient Fine-Tuning"
_ICLR.cc/2026/Conference — ICLR 2026 Conference Withdrawn Submission_

### Official Review · Reviewer_GgE5 · 2025-10-24

**Soundness:** 1
**Presentation:** 2
**Contribution:** 1
**Rating:** 2
**Confidence:** 4

**Summary:**

This paper proposes NoLoRA, a nonlinear extension of LoRA that introduces an activation function and a learnable modulation vector between the low-rank matrices A and B. The claimed motivation is to overcome the linearity limitation of LoRA and to improve representational capacity while maintaining parameter efficiency. Experiments show improvements across a bunch of benchmarks.

**Strengths:**

- The paper is clearly written and easy to follow.


- The method is straightforward, with a simple addition of a nonlinear activation and a per-rank modulation vector.


- The empirical section covers several benchmarks and is rigorous.

**Weaknesses:**

`W1: Novelty concerns and missing citation of highly related prior work`

The core technical contribution, introducing a nonlinear activation between the two LoRA matrices, is not new. The AuroRA paper (https://arxiv.org/abs/2505.18738) presented an almost identical idea several months earlier: inserting a nonlinear mapping between (A) and (B) to enhance LoRA’s expressiveness, effectively treating the adapter as a miniature MLP.

In fact, NoLoRA’s formulation $ \Delta W(x) = B (v \odot f(Ax))$ reduces to AuroRA’s nonlinear adapter when the modulation vector (v) is removed. Table 6 of this submission even includes an ablation explicitly without (v), which is practically equivalent to AuroRA.

Yet, the paper does not cite AuroRA anywhere in the related work or discussion. This omission gives a misleading impression of originality and fails to situate the contribution in its proper research context.

`W2: Lack of conceptual novelty beyond a modulation term`

Once the nonlinearity is recognized as prior art, the only remaining addition is the elementwise modulation vector (v), which adds minimal expressiveness and negligible theoretical depth. The proposed update remains a trivial per-channel scaling of the activation output. This does not constitute a fundamentally new idea or mechanism.

`W3: No justification or insights provided`

The paper does not provide any solid theoretical or empirical argument explaining why the method works well. On line 234, the authors state:
“This analysis illustrates the improved expressiveness of our method and provides theoretical support for the empirical results.”

However, no such analysis or theoretical support is actually presented.

`W4: Experimental validation lacks rigor`

The experiments do not include AuroRA for comparison, even though it is the most relevant prior method. Moreover, the improvements attributed to the modulation vector are marginal, casting doubt on the significance of this component.

---

While the paper is well written and includes broad experiments, its main technical idea is essentially identical to previously published work (AuroRA), minus proper attribution. The remaining addition, a simple modulation vector, is minor and not conceptually sufficient to justify a new standalone paper.

**Questions:**

Please refer to weaknesses

---

### Official Review · Reviewer_wkF1 · 2025-10-26

**Soundness:** 2
**Presentation:** 2
**Contribution:** 1
**Rating:** 2
**Confidence:** 4

**Summary:**

This manuscript focuses on low-rank adaptation (LoRA), which suffers from limited effectiveness due to its linear adapter architecture. To overcome this expressiveness bottleneck, this paper advocates a nonlinear variant termed Nonlinear Low-Rank Adaptation (NoLoRA) that injects a nonlinearity and a vector modulation between the low-rank adapters to enhance the representational capacity. Experiments are conducted on commonsense reasoning, natural language understanding, image classification, and mathematical reasoning to demonstrate the superiority of NoLoRA.

**Strengths:**

1. LoRA is a highly popular and timely topic in parameter-efficient fine-tuning. The limitation of LoRA's linear structure is clearly presented.
2. Empirical evaluation on diverse tasks and models showcase promising results.
3. NoLoRA is lightweight, incurring negligible extra parameters.

**Weaknesses:**

1. The motivation behind the specific design in Eq. (6) is not clearly explained, and there is no theoretical justification supporting the claimed improvement in expressiveness.
2. The comparison omits several closely related LoRA variants that also incorporate nonlinear structures. For instance, MoRA [1] replaces linear mappings with compression and decompression functions, while HiRA [2] employs a Hadamard product with pretrained weights.
3. The experimental results lack error bars (e.g., standard deviation or confidence intervals) and do not report performance across multiple random runs.
4. While Table 4 compares the parameter counts of different approaches, it would also be informative to include measurements of actual fine-tuning time and memory overhead.
5. The first paragraph of Section 3.3 repeats similar sentences in the last paragraph of Section 3.2. Additionally, “Mixture” in line 59 should be “mix,” and there should be a space before “NEAT” in line 107.

[1] T. Jiang et al., "Mora: High-rank updating for parameter-efficient fine-tuning", arXiv preprint, 2024.
[2] Q. Huang et al., "HiRA: Parameter-efficient hadamard high-rank adaptation for large language models", in ICLR, 2025.

**Questions:**

See weakness.

---

### Official Review · Reviewer_cKep · 2025-10-29

**Soundness:** 1
**Presentation:** 3
**Contribution:** 1
**Rating:** 2
**Confidence:** 4

**Summary:**

The paper proposes NoLoRA (Nonlinear Low-Rank Adaptation), a parameter-efficient fine-tuning (PEFT) method that extends LoRA by introducing a nonlinear activation function and a learnable modulation vector into the low-rank update path. The authors claim this design enhances expressiveness while preserving parameter efficiency.

**Strengths:**

1.	The core idea—enhancing LoRA with lightweight nonlinearity—is simple and aligns with the PEFT community’s goal of improving expressivity without sacrificing efficiency.
2.	The empirical scope is broad, covering NLP, vision, and reasoning tasks, which suggests general applicability.

**Weaknesses:**

1.	Lack of novelty: The proposal is extremely close to NEAT (Zhong et al., 2025). Both methods replace LoRA’s linear update with a nonlinear mapping. The paper fails to justify why this form is preferable or meaningfully distinct.
2.	Unverified experimental claims: Most baseline numbers are borrowed from other papers with different settings (e.g., learning rates, seeds, data splits). For example, the GLUE results for LoRA and Adapter are marked as taken from Wu et al. (2024a). Without re-running all baselines under identical conditions, the reported gains may reflect implementation or tuning disparities, not intrinsic superiority.
3.	Theoretical claims are vague: The statement that “for any smooth target weight update ΔW , there exists a set of parameters A,B,v such that B(v⊙f(Ax)) can approximate ΔW to arbitrary precision” is unsubstantiated. This would require f to be a universal approximator, but with fixed low rank r, the expressivity is severely limited.

**Questions:**

1.	What exactly makes NoLora better than NEAT? Any theoretical explanations?
2.	Experimental fairness: Were all baselines (LoRA, PiSSA, MiLoRA, NEAT, etc.) re-implemented and tuned under identical conditions (same seeds, hyperparameters, data preprocessing)? If not, how can the performance gaps be attributed to architectural differences rather than tuning disparities?

---

### Note · Authors · 2026-01-05

I have read and agree with the venue's withdrawal policy on behalf of myself and my co-authors.